# Economic Growth Targets and Carbon Emissions: Evidence from China

**DOI:** 10.3390/ijerph19138053

**Published:** 2022-06-30

**Authors:** Keliang Wang, Bin Zhao, Tianzheng Fan, Jinning Zhang

**Affiliations:** 1School of Economics, Ocean University of China, Qingdao 266100, China; wkl@ouc.edu.cn (K.W.); zhaobin9857@stu.ouc.edu.cn (B.Z.); 2School of Economics and Management, Xinjiang University, Urumqi 830046, China; 3School of Business, Shandong University, Weihai 264209, China; jnzhang003@mail.sdu.edu.cn

**Keywords:** economic growth target, resource misallocation, industrial restructuring, carbon emissions

## Abstract

Carbon emissions have become a new threat to sustainable development in China, and local government actions can play an important role in energy conservation and emission reduction. This paper explores the theoretical mechanisms and transmission paths of economic growth targets affecting carbon emissions from the perspective of economic growth targets and conducts an empirical analysis based on 30 provincial panel data in China from 2003 to 2019. The results show that: economic growth targets are positively correlated with carbon emissions under a series of endogeneity and robustness; there are regional heterogeneity, target heterogeneity and structural heterogeneity in the impact of economic growth targets on carbon emissions; after economic growth targets are set, government actions can influence carbon emissions by affecting resource mismatch and industrial restructuring; It is further found that there is a “U” shaped relationship between economic pressure and carbon emissions. Based on the above findings, this paper further proposes that a high-quality performance assessment mechanism should be developed to bring into play the active role of local governments in achieving carbon reduction goals, and thus contribute to high-quality economic development.

## 1. Introduction

Economic growth targets (EGTs) have become an important phenomenon in the economic growth process all over the world. Since 1950, at least 49 economies worldwide have announced economic growth targets, including developed countries such as Germany, the United Kingdom, Japan, and South Korea, as well as developing countries such as China and India [1,2,3]. Among these countries, China has the highest enthusiasm for setting goals. Chinese governments at all levels have announced their economic growth goals in the reports of the Party Congress, the development planning outline and the government work report, and hope that local governments will achieve them steadily [4]. Since China’s reform and opening up, the Chinese government has found that the economic growth target is an effective policy tool to achieve economic growth. However, many scholars have noticed that the setting of economic growth goals will bring about serious environmental problems [5,6]. In order to achieve the target, the local government intervenes in the allocation of resources within its jurisdiction and introduces heavily polluting industries, resulting in the wanton discharge of pollutants. Governments even relax environmental regulation to compete for the entry of these enterprises, causing the bottom effect of environmental regulation [7,8]. The vulnerability of the whole ecosystem is increasing under such a system. It is necessary for this paper to explore the relationship between economic growth goals and the environment. Especially under the political system of vertical management in China, the appointment and removal of local government officials is decided by the central government [9]. Local officials of lower-level governments tend to set higher economic growth targets than those of higher-level governments in order to enable higher-level governments to observe their efforts and ensure their smooth promotion. However, due to China’s vertical political system, local officials at lower levels of government tend to set higher economic growth targets than those at higher levels of government in order to have their efforts observed by higher levels of government and ensure their smooth promotion. As a result, with the cascading of economic growth targets at all levels of government, the pressure on provincial governments for economic development is great [10,11], which will have a more significant impact on the ecological environment of the jurisdiction.

As a global energy producer and consumer, China plays a vital role in global carbon emission reduction. With China’s economic development entering the fast track after the reform and opening up, the long-standing development approach of high input and high pollution has led to a surge in carbon emissions [12,13,14], the world share of carbon emissions rose from 16.88% in 2003 to 29.69% in 2018 (The data comes from EPS database. The website is: http://olap.epsnet.com.cn/auth/platform.html?sid=6A848C9509AEC576552E9B576C37190C, accessed on 18 June 2022). If the Chinese government does not take any measures to reduce carbon emissions, China’s carbon emissions will continue to grow, causing irreversible damage to the entire ecological environment. Governments around the world are aware of the danger of carbon emissions and have introduced corresponding carbon reduction policies, such as the U.S. proposal to achieve “Net-Zero Carbon” by 2050, and the EU government has submitted the European Climate Law to legally address the issue of carbon emissions. Similarly, as a responsible country, the Chinese government has been actively dealing with the issue of carbon dioxide emissions [15,16]. Then, as a policy tool of governments at all levels, the economic growth target is bound to have an important impact on carbon emissions. Although the economic growth target is declining with the adjustment of China’s industrial structure and the gradual disappearance of labor advantages, this assessment of economic development still affects the behavior of the government. Therefore, this paper clarifies the impact of economic growth goals on carbon emissions and its mechanism, which has important theoretical significance for enriching the relevant theories of sustainable development and practical significance for realizing carbon emission reduction.

The marginal contributions of this paper are mainly in the following four aspects: (1) In terms of research perspective, this paper explores the direct impact of economic growth targets on carbon emissions and introduces resource mismatch and industrial restructuring as the mechanisms of the impact of economic growth targets on carbon emissions. (2) In terms of heterogeneity, we investigate the impact of economic growth targets on carbon emissions based on regional heterogeneity, heterogeneity of economic growth targets and heterogeneity of industrial weight. In further investigation, this paper introduces economic growth pressure to explore the nonlinear effects of economic growth pressure on carbon emissions. (3) In terms of research significance, the study of carbon emissions from the perspective of economic growth targets is not only a relevant supplement to the existing studies, but also a macro-level interpretation of the impact of economic growth targets on carbon emissions, which is important for the future practice of low-carbon development, achieving the “double carbon” target, better defining the functions and roles of the government, and promoting the high-quality economic development of China.

The rest of the paper is organized as follows. Part II reviews the relevant literature and proposes research hypotheses. Part III describes the methodology and data used to conduct the empirical study. Part IV provides the empirical results, including baseline regression analysis, heterogeneity analysis, and further analysis. Part V summarizes the conclusions of the paper and provides corresponding policy recommendations.

## 2. Literature Review and Research Hypotheses

### 2.1. Relevant Research on Carbon Emission

Practicing low-carbon development and achieving the “double carbon” goal have become hot topics of academic research. Through a review of the literature, the relevant research mainly focuses on the measurement of carbon emissions and the impact factors of carbon emissions. For the measurement of China’s carbon emissions, the mainstream method is based on the carbon emission factors provided by IPCC, for example, Chi et al. [17] and Chen et al. [18] estimate carbon emissions at the provincial level in China, and Duren and Miller [19] measure carbon emissions in the world’s megacities and assess future trends. For the study of factors influencing carbon emissions, existing studies can be divided into two main aspects, namely market economy factors and governmental factors. The market economy factors include urbanization [20,21], rapid economic development [22] and other factors, resulting in an industrial structure characterized by energy-intensive and high-polluting heavy industries, which leads to a surge in carbon emissions. It has also been argued that the market-based emissions trading mechanism and sewage charges have not exerted their positive effects on the ecological environment [23], so the institutional arrangements of the government should not be ignored, and studies have shown that pollution monitoring and administrative control are the main reasons for the improvement of environmental quality in Western developed countries [24]. In the context of China’s reality, administrative intervention by the government is an important tool for environmental governance, which includes environmental regulatory tools for local governments [25], the design of officials’ assessment represented by the central government [26], and institutional arrangements such as fiscal decentralization under political centralization [27].

### 2.2. Direct Impact of Economic Growth Goals on Carbon Emissions

Local government officials in China are appointed by the central government. In order to enable the superior government to observe their own efforts, local government officials tend to set higher economic growth goals than the superior government [4,28,29]. Driven by this economic tournament, local governments will launch fierce economic competition. Many scholars have found that under the economic competition model, when environmental protection and economic development goals are incompatible, local governments prefer to improve economic performance and use their power to allocate resources to economic development [2,30]. Moreover, in order to attract domestic and foreign enterprises to invest and create more jobs in their jurisdictions, local governments will lower environmental regulation standards, relax environmental regulations, and engage in environmental “race to the bottom” competition, increasing carbon emissions [31,32]. Xu and Gao [4] found that the economic growth target is the central government’s assessment of the local government’s economic development achievements in that year, so local officials inevitably have short-sighted behavior. Attracting heavy industrial enterprises with high pollution and high output into the jurisdiction has become an important way to achieve the economic growth goal, thus deteriorating the ecological environment [33,34]. On this basis, Dong [35] found that local governments’ enthusiasm for heavy industrial enterprises would potentially lead social funds to flow to these fields, resulting in overheated investment in heavy industrial enterprises, which further promotes the generation of carbon emissions. In addition, due to the long cycle and unstable return of innovation investment, it is difficult for long-cycle innovative investments to enter the utility function of local officials, which often leads to the status quo of “production over innovation” in the jurisdictions and is detrimental to the improvement of regional innovation levels, resulting in innovation crowding-out effects [30,36,37].

### 2.3. Indirect Impact of Economic Growth Goals on Carbon Emissions

Economic growth targets not only have direct effects on carbon emissions but also have indirect effects on carbon emissions through resource mismatch and industrial restructuring, which are mainly reflected in the following two paths.

First, the resource mismatch path. Liu et al. [30] theoretically proved that under certain conditions, the economic growth goals set by local governments can affect economic growth through forced resource allocation. Under the GDP-based assessment mechanism, local governments will allocate their own resources and guide social resources to projects that meet their economic growth goals [38]. It has become an effective policy tool choice for local governments by intervening in the allocation of factor resources and distorting factor prices to promote rapid economic growth in their jurisdictions [30,39]. However, due to the imperfect market economy system and government administrative intervention, some industrial sectors are overinvested and have serious overcapacity [40,41,42], distorting the normal resource allocation results of the market. At the same time, the allocation of resource factors is further distorted by the “cascading” of economic growth targets by local governments [43], leading to an increase in carbon emissions. In addition, local governments restrict the cross-regional flow of resources, factors, and products to safeguard their own interests, leading to market segmentation. Governments at all levels often adopt local protection policies for the economy within their jurisdictions in order to accomplish their expected economic growth targets, which makes some enterprises face size constraints and hinders the improvement of factor allocation efficiency [44,45]. It has been shown that the GDP-based performance evaluation mechanism leads to local officials’ efforts to favor economic growth, resulting in distorted resource allocation and higher carbon emissions [46].

Second, the path of industrial restructuring. Late developing countries promote industrial restructuring and upgrading through industrial policies, however, in reality, the effect of policy formulation and implementation depends largely on government behavior [33]. Under the GDP-centered economic performance assessment, once there is a change in local government, the new officials in the jurisdiction need to make more excellent achievements than their predecessors in order to obtain a greater chance of promotion capital. As a result, new officials often appear to negate the industrial policies formulated by their predecessors [47], resulting in the government’s implementation of industrial policies that often deviate from the policy objectives. Large amounts of resources flow in and out frequently within different industries [48], constraining the optimization and upgrading of regional industrial structures. It has been shown that industrial restructuring and upgrading is an important factor in developing a low-carbon economy [49,50,51], while Gu et al. [52] further showed that industrial restructuring and upgrading is an effective driver of CO_2_ emission reduction, as well as a fundamental driver of the low-carbon development of the industrial system. Based on the theoretical analysis of the above scholars, this paper draws the path map of the impact of economic growth goals on carbon emissions, and puts forward hypotheses 1 and 2:
**Hypothesis** **1.***Economic growth targets may exacerbate carbon emissions*.
**Hypothesis** **2.***Economic growth targets can trigger resource misallocation and constrain industrial restructuring and thus act on carbon emissions*.

The roadmap of the theoretical mechanism is shown in Figure 1.

## 3. Methodology and Data

### 3.1. Model Settings

First, the following model is constructed to explore the mechanism of the direct effect of economic growth targets on carbon emissions, as shown in Model (1).
(1)$$AC_{it}=\alpha_{0}+\alpha_{1}EGT_{it}+\overset{k}{\underset{n=2}{\sum }}\alpha_{n}X_{it}+\varepsilon_{it}$$
*i* and *t* denote region and year, respectively, *AC* denotes carbon emissions per capita, *EGT* is economic growth target and *X* indicates control variables. α denotes the correlation coefficient and *ε_it_* is the random error term.

Second, to explore the indirect mechanism of the effect of economic growth targets on carbon emissions, this paper draws on the research method of Zhang et al. [53] and uses a mediating effect model for empirical testing, which is as follows.
(2)$$RES_{it}=\beta_{0}+\beta_{1}EGT_{it}+\sum_{n=2}^{k}\beta_{n}X_{it}+\varepsilon_{it}$$
(3)$$AC_{it}=\chi_{0}+\chi_{1}EGT_{it}+\chi_{2}RES_{it}+\overset{k}{\underset{n=3}{\sum }}\chi_{n}X_{it}+\varepsilon_{it}$$
(4)$$IND_{it}=\delta_{0}+\delta_{1}EGT_{it}+\overset{k}{\underset{n=2}{\sum }}\delta_{n}X_{it}+\varepsilon_{it}$$
(5)$$AC_{it}=\gamma_{0}+\gamma_{1}EGT_{it}+\gamma_{2}IND_{it}+\overset{k}{\underset{n=3}{\sum }}\gamma_{n}X_{it}+\varepsilon_{it}$$

*RES* and *IND* denote resource misallocation and industrial structure upgrading, respectively. *β*, *χ*, *δ*, and *γ* are the regression coefficients of each variable.

Model (2), model (3) and model (1) are the mediating effect models in which economic growth targets affect resource misallocation and then act on carbon emissions. Model (4), model (5) and model (1) are the mediating effect models in which the economic growth target affects the upgrading of industrial structure and then acts on carbon emissions.

### 3.2. Variable Description and Data Sources

The explained variable, core explanatory variable, mediating variables, and control variables are treated specifically as follows.

#### 3.2.1. Explained Variable

Carbon emissions per capita (AC). Along with the acceleration of global industrialization and urbanization, global energy consumption, especially the consumption of traditional fossil energy, will increase year by year, further leading to the rising total CO_2_ emissions [54,55]. Based on the carbon emission coefficients published by IPCC and the actual situation in China, the consumption of eight fossil fuels, namely diesel, coal, coke, gasoline, kerosene, fuel oil, natural gas and crude oil, which are closely related to carbon emissions, is selected to estimate the total carbon emissions. The relevant calculation formula is shown in Equations (6) and (7), and the calculated total carbon emissions are divided by the total regional population to calculate the per capita carbon emissions as the explanatory variables in this paper.
(6)$$CO_{2}=\overset{H}{\underset{i=1}{\sum }}E_{i}\times CEF_{i}$$
(7)$$CEF_{i}=H_{i}\times CH_{i}\times COR_{i}\times CE_{i}\times \frac{44}{12}\times 10^{-6}$$

*E_i_* represents the total consumption of the energy *i*, *CEF_i_* indicates the carbon emission factor of the energy *i*. *H_i_*, *CH_i_*, *COR_i_* denote the average low-level heat generation, carbon content per unit calorific value, carbon oxidation rate and final carbon emission factor, respectively. The interpretation of the remaining indicators is shown in Table 1.

#### 3.2.2. Core Explanatory Variable

Economic growth target (EGT). Drawing on the treatment of Shen [36] and Liu et al. [30], the economic growth rate targets announced in the government work report of each province are chosen to measure the economic growth target.

#### 3.2.3. Mediating Variables

Resource mismatch (RES). Resource mismatch between regions not only leads to inefficient resource allocation but also affects the way the economy mixes output in the long run. Therefore, referring to the research method of Hao et al. [56], the capital mismatch index (CAP) and labor mismatch index (LAB) are used to measure the degree of resource mismatch in each province and region, respectively.

We assume that a factor market has a distorted competitive market [57], and defines the absolute distortion coefficients of capital and labor factors as Formula (8):(8)$$\gamma_{CAP}=\frac{1}{1+\tau_{CAPi}},\;\gamma_{LAB}=\frac{1}{1+\tau_{LABi}}$$
where, $\gamma_{CAP}$ and $\gamma_{LAB}$ are absolute distortion coefficients of factor prices, indicating that resources have relatively no distorted bonus. In the actual measurement, the relative distortion coefficient can also be used to replace it.
(9)$${\hat{\gamma }}_{CAPi}=(\frac{CAP_{i}}{CAP})/(\frac{s_{i}\beta_{CAPi}}{\beta_{CAP}}),\;{\hat{\gamma }}_{LABi}=(\frac{LAB_{i}}{LAB})/(\frac{s_{i}\beta_{LABi}}{\beta_{CAP}})$$
where, $s_{i}=\frac{p_{i}y_{i}}{Y}$ represents the share of regional output in the overall economic output, and $\beta_{CAP}=\sum_{i}^{N}s_{i}\beta_{CAPi}$ represents the output weighted capital contribution value. $\frac{CAP_{i}}{CAP}$ represents the actual proportion of regional capital used in the total capital, and $\frac{s_{i}\beta_{CAPi}}{\beta_{CAP}}$ is the theoretical proportion of regional capital used in the effective allocation of capital. If we want to further calculate the capital mismatch index and labor mismatch index, we need to calculate the factor output elasticity of capital and labor in each region. We assume that the production function is a C-D production function with constant returns to scale.
(10)$$Y_{it}=ACAP_{it}^{\beta_{CAPi}}LAB_{it}^{1-\beta_{CAPi}}$$

Both sides take logarithms at the same time, and add individual effects and time effects to the model, specifically:(11)ln(YitLABit)=lnA+βCAPiln(CAPitLABit)+μi+λt+εit

After estimating the factor output elasticity of each province, the resource mismatch index and labor mismatch index of each province are calculated according to (8) and (9).

Industrial structure upgrading (IND). In order to comprehensively reflect the connotation of industrial structure upgrading, this paper draws on the research method of Wu et al. [55] and takes into account the primary industry, secondary industry and tertiary industry at the same time to construct the industrial structure upgrading index, which is measured as shown in Equation (12).
(12)$$IND_{it}=\sum_{s=1}^{3}x_{s}\times s,\;\;1\leq IND\leq 3$$

*IND* denotes the industrial upgrading index of province *i* in year *t*, and *x_s_* denotes the share of the three major industries in the total economy, respectively.

#### 3.2.4. Control Variables

Referring to the existing studies [10,36,58], the following control variables are selected in this paper. The Innovation level (INN), measured by the number of patent applications granted in each province. The Urbanization level (UR), measured by the urban population to total population in each province. The foreign direct investment level (FDI), measured using the amount of foreign direct investment and converted to RMB based on annual exchange rates. Transportation infrastructure (TAR), which is measured using private vehicle ownership as the vast majority of CO_2_ emissions come from energy consumption in industry and automobiles. Government Intervention (GOV), measured using the share of government fiscal spending in the total economy in each province. Secondary industry (SIND), measured using the share of secondary industry in the total economy.

#### 3.2.5. Data Sources

This paper interprets the impact of economic growth targets on carbon emissions based on a panel of 30 provinces in China from 2003 to 2019 (Hong Kong, Macau, Taiwan and Tibet are not considered in this paper because of certain data deficiencies). The data on economic growth targets are obtained from provincial government work reports, the data on carbon emissions are obtained from the *China Energy Statistics Yearbook*, and other raw data are obtained from the *China Statistical Yearbook*, the CEI statistical database, and the statistical yearbooks of each province. The descriptive statistics of specific variables are shown in Table 2.

## 4. Empirical Results

### 4.1. Benchmark Regression Analysis

First, in order to interpret the direct impact of economic growth targets on carbon emissions, this paper uses a static panel for regression, where column (1) in Table 3 is the regression result of the panel fixed-effects model and column (2) is the regression result of the panel random-effects model. The Hausman test shows that the regression results of the fixed-effects model outperform the regression results of the random-effects model, so the regression analysis is conducted for column (1) in Table 3. When other variables are held constant, economic growth targets and carbon emissions show a positive correlation, i.e., the higher the economic growth target is set, the higher the carbon emissions will emit. This also requires regional governments to take into account regional resource endowment, economic development level and ecological and environmental conditions when setting economic growth targets, and to build a system of a virtuous cycle between economic growth and carbon emissions. Among the control variables, the improvement of regional innovation level and the increase in foreign investment introduction will be beneficial to mitigate carbon emissions. By comparing the relevant regression coefficients, we can find that the improvement of regional innovation levels is the key to reducing carbon emissions. Along with the improvement of China’s business environment, the government should pay more attention to the quality of foreign investment to better exploit the technology spillover effect and mitigate carbon emissions. The introduction of foreign investment in China is changing from a pollution paradise to a pollution halo. The increase in urbanization level, the improvement of transportation facilities, the increase in government intervention and the development of industry all increase carbon emissions, and it can be seen that the factors causing the increase in carbon emissions are diversified, so in the process of future development, more attention should be paid to the following points. First, in the process of urbanization, the government should make the urbanized population, industry and land match each other to avoid the inefficient allocation of resources caused by aggressive urbanization. Second, the government should encourage green travel and increase support for the field of new energy vehicles. Third, the government needs to optimize the fiscal expenditure structure. Fourth, the government should improve the carbon emission efficiency of traditional industries such as high energy consumption and high pollution with the help of technological innovation and a carbon market trading system.

Furthermore, considering the possible two-way causality of endogeneity between economic growth targets and carbon emissions, the instrumental variable approach is used to test the endogeneity problem. Specifically, the economic growth target is treated as an instrumental variable for the economic growth target with both a one-period lag and a two-period lag. In addition, local governments use the previous year’s actual economic growth as a reference to set the current year’s economic growth target, so the paper also uses the previous year’s actual economic growth as the instrumental variable for the economic growth target. The specific regression results are shown in columns (3) and (4) in Table 3. The KP-LM statistics of both methods above pass the significance test in the first stage, as well as the CD-Wald F statistic are significantly larger than the critical value of 16.39 calculated by Stock and Yogo [59] for the F value at a 10% bias level, indicating that there is no weak instrumental variable problem. The regression coefficients all pass the significance test, again indicating that economic growth targets and carbon emissions show a positive relationship.

Finally, in order to test the reliability and robustness of the regression results, this paper uses substitution of the relevant variables before regression. Specifically, it includes the following three aspects. (1) The explained variable is replaced. Carbon emissions are replaced with industrial sulfur dioxide emissions (SO_2_), and the regression results are shown in column (5) of Table 3. (2) The core explanatory variable is replaced. The current year’s economic growth target is replaced by the current year’s actual economic growth (AEG), and the regression results are shown in column (6) of Table 3. (3) The control variables are replaced. In this paper, we change the measurement of regional innovation by replacing the number of patent applications granted with the comprehensive utility value of regional innovation capacity (UT). The data are obtained from the *China Regional Innovation Capacity Evaluation Report*, and the regression results are shown in column (7) of Table 3. After comparing the regression results in columns (5), (6) and (7) with column (1), it can be seen that there is no significant change in the sign direction as well as the significance of the core explanatory variables, indicating the robustness and reliability of the baseline regression results.

### 4.2. Heterogeneity Analysis

Regarding the heterogeneity exploration, this paper will explore the impact of economic growth targets on carbon emissions from three aspects: regional heterogeneity, heterogeneity of the difference between economic growth targets and actual economic growth, and industrial heterogeneity. First, regarding the regional heterogeneity, the full sample is divided into the eastern region and the central and western regions, and this division is based on the fact that the total economic volume of the eastern region and the central and western regions are roughly comparable, and the specific regression results are shown in Table 4. The regression coefficients of economic growth targets in Table 4 show that the higher the economic growth target in the eastern region, the greater the impact on total carbon emissions compared to the central and western regions. The relatively economically developed eastern region has relatively well-developed industrial and infrastructure conditions and is also a key region for GDP performance assessment. Local governments tend to set higher economic growth targets and introduce various policies to promote early or even over-achievement of economic growth targets in their jurisdictions, and tend to adopt short-term crude economic behavior [60,61]. Therefore, the increase in economic growth targets in the eastern region is more likely to promote carbon emissions. Local governments in the central and western regions have gradually taken over the transfer of heavily polluting industries from the eastern regions in order to achieve their economic growth targets, which has also led to an increase in carbon emission pressure in the region.

Second, economic growth target heterogeneity. In this paper, we use the difference between the current year’s economic growth target and the actual economic growth to investigate the impact of economic growth targets on carbon emissions when the economic growth target is higher than the actual economic growth or when the economic growth target is lower than the actual economic growth, and the specific regression results are shown in Table 5. As can be seen from Table 5, under the premise that other variables remain unchanged, whether the economic growth target is higher or lower than the actual economic growth, the economic growth target and carbon emission still show a positive correlation, and the regression coefficients are basically the same. It indicates that economic development and carbon emission reduction are incompatible, regardless of whether the economic growth target is exceeded in the current year. To resolve this contradiction, the Chinese government requires that economic growth targets be set in a reasonable range and adjusted as appropriate from year to year, as well as a clear vision for energy conservation and carbon reduction.

Finally, the heterogeneity of industry composition. Local governments’ preference for short-term investment strategies to achieve economic growth goals will gradually be reflected at the level of the dominant industries in their jurisdictions [60]. In the case of the secondary industry, its development is in line with local governments’ intention to achieve economic growth in the short term, and therefore the secondary industry has become the focus of local governments’ attention [62]. Therefore, based on its idea, we classify the share of the secondary industry in the economy as whether it is higher than 50% and further analyze the impact of economic growth targets on carbon emissions, and the regression results are shown in Table 6. As can be seen from Table 6, when the share of the secondary industry in the total economy is higher than 50%, the impact of the economic growth target on carbon emission shows a significant positive relationship. In regions with a high proportion of the secondary industry, it indicates that local governments are more inclined to invest in an industrial-based secondary industry with a short cycle time and quick results in order to accomplish the economic growth target of the year, which causes distortion of industrial structure and duplication of construction. In the context of an irrational industrial structure, the economic growth target leads to a serious waste of resources, which is not conducive to mitigating regional carbon emissions.

### 4.3. Analysis of the Intermediary Effect

The above theoretical mechanism shows that economic growth targets can affect carbon emissions by influencing resource mismatch and industrial structure upgrading, and the specific regression results are shown in Table 7. As shown in Table 7, the higher the degree of capital mismatch and labor mismatch, the less conducive to carbon emission mitigation. On the one hand, restrictions by local governments such as the household registration management system and the land property rights trading system lead to labor market segmentation, which is not conducive to the free flow of labor and hinders the knowledge spillover effect of talents [63]. On the other hand, local governments tend to invest limited capital in state-owned enterprises, resulting in excessive investment and duplication of construction. The misallocation of social resources reduces economic efficiency and increases carbon emissions.

It is also evident from Table 7 that the increase in economic growth targets hinders the optimization and upgrading of industrial structure, however, the optimization and upgrading of industrial structure are beneficial to mitigate carbon emissions. It shows that local governments often make direct administrative intervention in industrial structure adjustment to achieve economic growth targets, which makes local industrial structure upgrading blunt [64]. Therefore, the government should take into account the differences in resource endowment, development stage and economic environment of each region to determine different economic growth targets, so as to achieve optimization and upgrading of industrial structure and reduction in carbon emissions.

### 4.4. Further Analysis

In the GDP performance appraisal system, the central government forms important incentives for local officials on the one hand and puts economic growth pressure on local officials on the other hand. Therefore, under different economic growth pressures, local government officials adopt different policy intensities and paces [65], so does the effect of economic growth pressure on carbon emissions show a simple linear or nonlinear relationship? To answer this question, this paper draws on Zhu and Lin [66] to measure economic growth pressure (PRE) using the ratio of the economic growth rate target of the year to the actual economic growth rate in the previous year. The core explanatory variables in model (1) are replaced with the squared terms of economic growth pressure and economic growth pressure, and the other control variables remain unchanged. To further corroborate the reliability and robustness of the regression results, the full sample is further divided into eastern and central-western regions for heterogeneity exploration, and the regression results are shown in Table 8. As can be seen from Table 8, both for the full sample and regional heterogeneity, the pressure of economic growth shows a “U” shaped relationship on carbon emissions, indicating that moderate pressure of economic growth has a suppressive effect on carbon emissions, but too high pressure of economic growth may distort the economic policy tendency of local governments and officials and may drive local governments to adopt a sloppy development model that “emphasizes the economy at the expense of the environment” and even neglect environmental management. The imbalance in the relationship between economic construction and environmental protection [67,68] adversely affects economic efficiency and the reduction of carbon emissions.

## 5. Conclusions

This paper explores the impact of economic growth targets on carbon emissions based on 30 provincial panel data in China from 2003–2019. The study shows that (1) with full consideration of endogeneity and robustness, economic growth targets show a positive relationship with carbon emissions. (2) Heterogeneity exploration shows that, firstly, the increase in economic growth targets in eastern, central and western regions promotes carbon emissions, and the impact of economic growth targets on carbon emissions is higher in eastern regions than in central and western regions. Secondly, there is an incompatibility between economic growth and mitigation of carbon emissions regardless of whether the government can exceed the economic growth targets. Finally, when the output value of the secondary industry accounts for more than 50% of the total economy, the achievement of economic growth targets increases regional carbon emissions. (3) The transmission mechanism suggests that the local government’s behavior under the economic growth target will trigger resource misallocation and industrial restructuring, and thus affect the carbon emission level. (4) Further analysis reveals that there is a non-linear “U” shaped relationship between economic growth pressure and carbon emissions. The findings of the study further support that, while economic growth has achieved remarkable performance, the negative externalities such as overcapacity and environmental pollution brought about by the unilateral pursuit of economic growth rate are constantly highlighted, and too fast an economic growth rate may erode the quality of economic growth.

Based on the above conclusions, we make the following recommendations to achieve a harmonious coexistence of economic development and carbon emission reduction, and thus contribute to China’s high-quality economic development.

First, the central government should enrich and optimize the content and structure of the performance appraisal of local officials to avoid local officials from destroying the environment for the sake of promotion. At the same time, local governments should set moderate economic growth targets taking into account local resource endowment, development stage and economic environment, avoid setting “too high” economic growth targets and abandon the past preference of local governments for “short and quick” projects.

Second, the government’s administrative means focus on combining with the market, reducing the government’s direct intervention in resources, playing an active role in resource allocation, and further improving the market-based carbon emissions trading system, while breaking the artificial market segmentation, promoting the free flow of factors, strengthening inter-regional cooperation and exchange, establishing joint prevention and control mechanisms, and forming a complementary pattern of advantages, with a view to achieving sustainable development.

There are still some deficiencies in this paper, mainly reflected in two aspects. On the one hand, this paper focuses on the impact of China’s provincial-level economic growth goals and carbon emissions, without an in-depth discussion of China’s urban level, which may lead to some deviation. Exploring the impact of economic growth goals on carbon emissions based on China’s urban level panel data is also a direction for future efforts. On the other hand, this study does not take into account the spatial interaction of economic activities. In fact, air pollution has a fluidity and transmission effect, and the practice of economic growth target management in a certain region is likely to affect the environment of adjacent regions. Therefore, it is expected to use the spatial econometric model to explore the spatial agglomeration and spatial spillover effects of economic growth goals on carbon emissions in the future.

## Figures and Tables

**Figure 1 ijerph-19-08053-f001:**
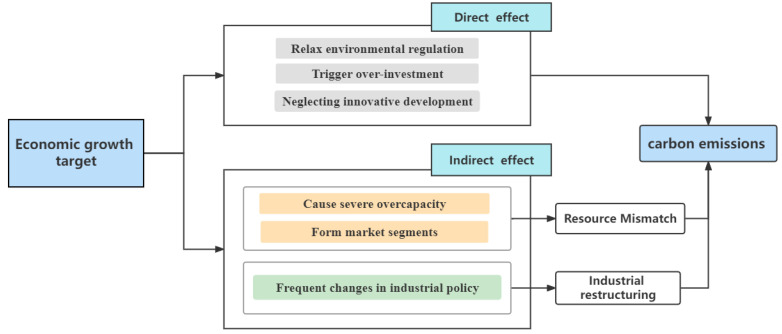
The roadmap of the theoretical mechanism.

**Table 1 ijerph-19-08053-t001:** Conversion table of carbon emission factors of major energy sources.

Energy Category	*H_i_* (KJ/KG)	*CH_i_* (TC/TJ)	*COR_i_*	*CEF_i_* (KGCO_2_/KG)
Diesel	42,652	20.2	0.98	3.10
Coke	28,435	29.5	0.93	2.86
Coal	20,908	26.4	0.94	1.90
Kerosene	43,070	19.5	0.98	3.02
Gasoline	43,070	18.9	0.98	2.93
Fuel oil	41,816	21.1	0.98	3.17
Natural gas	38,931	15.3	0.99	2.16
Crude oil	41,816	20.1	0.98	3.02

**Table 2 ijerph-19-08053-t002:** Descriptive statistics of variables.

Variables	Variable Symbols	Mean	Std.	Min	Max
Carbon emissions per capita	*AC*	9.257	7.997	1.415	52.223
Economic growth target	*EGT*	9.434	1.845	4.5	15
Capital mismatch	*CAP*	0.238	0.185	0.001	1.547
Labor mismatch	*LAB*	0.427	0.433	0.001	3.424
Industrial structure upgrading	*IND*	2.318	0.134	2.028	2.832
Innovation level	*INN*	32,430.24	60,524.14	70	52,7390
Level of Urbanization	*UR*	52.939	14.369	25.659	89.6
Foreign Investment Level	*FDI*	422.837	468.185	0.295	2257.322
Transportation Infrastructure	*TRA*	302.946	360.689	4.1	2092.39
Government Intervention	*GOV*	0.214	0.096	0.079	0.628
Secondary industry	*SIND*	45.730	8.339	16.2	61.5

**Table 3 ijerph-19-08053-t003:** The estimated results of the direct effect.

Variables	Explained Variable: lnAC (lnSO2)						
	Benchmark Regression Analysis		Instrumental Variables Regression		Robustness Regression		
	(1)	(2)	(3)	(4)	(5)	(6)	(7)
lnEGT (lnAEG)	0.142 **	0.122 **	0.103 *	0.352 ***	1.189 ***	0.145 ***	0.180 ***
	(0.056)	(0.057)	(0.054)	(0.074)	(0.152)	(0.037)	(0.059)
lnINN (lnUT)	−0.127 ***	−0.146 ***	−0.082 ***	−0.115 ***	−0.332 ***	−0.127 ***	−0.012
	(0.027)	(0.026)	(0.026)	(0.027)	(0.072)	(0.027)	(0.091)
lnUR	0.282 *	0.496 ***	−0.002	0.254	−0.371	0.231	0.121
	(0.154)	(0.140)	(0.163)	(0.155)	(0.415)	(0.153)	(0.160)
lnFDI	−0.022 *	−0.026 **	−0.031 ***	−0.032 **	−0.069 **	−0.022 *	−0.026 **
	(0.013)	(0.013)	(0.012)	(0.013)	(0.034)	(0.012)	(0.013)
lnTRA	0.336 ***	0.315 ***	0.287 ***	0.348 ***	0.124	0.362 ***	0.247 ***
	(0.033)	(0.032)	(0.035)	(0.033)	(0.089)	(0.034)	(0.028)
lnGOV	0.218 ***	0.278 ***	0.233 ***	0.180 **	0.884 ***	0.204 ***	0.163 **
	(0.077)	(0.069)	(0.072)	(0.078)	(0.206)	(0.076)	(0.078)
lnSIND	0.661 ***	0.673 ***	0.550 ***	0.465 ***	1.293 ***	0.638 ***	0.690 ***
	(0.084)	(0.083)	(0.083)	(0.095)	(0.226)	(0.076)	(0.086)
Constant	−2.004 ***	−2.455 ***			1.794	−1.894 ***	−2.314 ***
	(0.509)	(0.493)			(1.373)	(0.506)	(0.524)
KP-LM			416.791	281.759			
			[0.000]	[0.000]			
CD-Wald F			2.7 × 10^4^	674.251			
Observations	510	510	450	509	510	509	510
Number of id	30	30	30	30	30	30	30
R-squared	0.752	0.751	0.663	0.745	0.554	0.757	0.741

Notes: Standard errors in parentheses, *p*-values in square brackets, *** *p* < 0.01, ** *p* < 0.05, * *p* < 0.1.

**Table 4 ijerph-19-08053-t004:** Regression results of regional heterogeneity.

Variables	Explained Variable: lnAC	
	Eastern Region	Central and Western Region
lnEGT	0.295 ***	0.192 ***
	(0.077)	(0.074)
Control variables	YES	YES
Constant	−4.951 ***	0.875
	(0.752)	(0.760)
Observations	187	323
Number of id	11	19
R-squared	0.823	0.763

*** *p* < 0.01.

**Table 5 ijerph-19-08053-t005:** Heterogeneity regression results of economic growth targets.

Variables	Explained Variable: lnAC	
	Higher Than Real Economic Growth	Lower Than Real Economic Growth
lnEGT	0.215 **	0.212 ***
	(0.087)	(0.079)
Control variables	YES	YES
Constant	−1.268	−2.252 ***
	(1.577)	(0.590)
Observations	131	379
Number of id	30	30
R-squared	0.513	0.767

*** *p* < 0.01, ** *p* < 0.05.

**Table 6 ijerph-19-08053-t006:** Industry heterogeneity regression results.

Variables	Explained variable: lnAC	
	The Proportion of Secondary Industry Is Higher Than 50%	The Proportion of Secondary Industry Is Lower Than 50%
lnEGT	0.268 ***	0.071
	(0.084)	(0.075)
Control variables	YES	YES
Constant	−5.207 ***	−2.322 ***
	(1.092)	(0.701)
Observations	172	338
Number of id	22	30
R-squared	0.839	0.741

*** *p* < 0.01.

**Table 7 ijerph-19-08053-t007:** Intermediary effect results.

Variables	lnCAP	lnAC	lnLAB	lnAC	lnIND	lnAC
	(1)	(2)	(3)	(4)	(5)	(6)
lnEGT	−0.055	0.143 **	−0.138	0.149 ***	−0.022 ***	0.126 **
	(0.254)	(0.056)	(0.198)	(0.056)	(0.004)	(0.058)
lnCAP		0.026 **				
		(0.010)				
lnLAB				0.051 ***		
				(0.013)		
lnIND						−0.723
						(0.583)
Control variables	YES	YES	YES	YES	YES	YES
Constant	−1.174	−1.973 ***	−2.953 *	−1.852 ***	0.950 ***	−1.317 *
	(2.291)	(0.506)	(1.789)	(0.503)	(0.040)	(0.752)
Observations	510	510	510	510	510	510
Number of id	30	30	30	30	30	30
R-squared	0.070	0.756	0.047	0.760	0.867	0.753

*** *p* < 0.01, ** *p* < 0.05, * *p* < 0.1.

**Table 8 ijerph-19-08053-t008:** Regression results of economic growth pressure on carbon emissions.

Variables	Explained Variable: lnAC		
	Full Sample	Eastern Region	Central and Western Region
lnPRE	−0.213 ***	−0.087	−0.304 ***
	(0.059)	(0.067)	(0.082)
(lnPRE)^2	0.393 ***	0.065 ***	0.034 **
	(0.135)	(0.018)	(0.016)
Control variables	YES	YES	YES
Constant	−2.047 ***	−4.543 ***	0.919
	(0.500)	(0.763)	(0.761)
Observations	509	186	323
Number of id	30	11	19
R-squared	0.763	0.826	0.774

*** *p* < 0.01, ** *p* < 0.05.

## Data Availability

The data presented in this study are available on request from EPS database and the National Bureau of Statistics.

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
