# Peer review of "Economic Growth Targets and Carbon Emissions: Evidence from China"

_ijerph, 2022, doi:10.3390/ijerph19138053_

Round 1
Reviewer 1 Report
This subject's study is critical: this paper explores the theoretical mechanisms and transmission paths of economic growth targets affecting carbon emissions from the perspective of government intervention, and conducts an empirical analysis. Government actions can influence carbon emissions by affecting resource mismatch and industrial restructuring. There is a "U" shaped relationship between economic pressure and carbon emissions. However, there are still some problems in the article; here are my comments:
1. This article refers to a large number of literature about measurement of carbon emissions and the impact factors of carbon emissions. However, it is necessary to add the literature of the relationship between promotion of local government officials (competition for growth or competition for innovation), economic growth target constraint, innovation and environmental pollution.
2. The author's own theoretical contribution is not significantly reflected in this part.
3. It is necessary to demonstrate the measurement method of the capital mismatch index (CAP) and labor mismatch index (LAB).
4. What’s the relation between economic growth pressure, economic growth targets and carbon emissions. The authors should consider whether there is a threshold effect between economic growth pressure and carbon emissions.
5. Except for conclusions and policy recommendations, there is more room to focus on the limitations and research outlook.
Author Response
Dear Editors and Reviewers,
Thank you very much for carefully reading the manuscript. We are very grateful to your helpful and valuable recommendations for further improvements of the paper. We have revised the manuscript according to all of reviewers’ comments and suggestions. All the revised contents have been marked red in the manuscript, and the answers to the reviewers’ comments have been marked dark blue in this response letter. More details are as follows.
This article refers to a large number of literature about measurement of carbon emissions and the impact factors of carbon emissions. However, it is necessary to add the literature of the relationship between promotion of local government officials (competition for growth or competition for innovation), economic growth target constraint, innovation and environmental pollution.
R: Thank you very much for your valuable comments. We have added relevant documents on the relationships between local government competition and environmental pollution, economic growth goals and environmental pollution, as well as green technology innovation and pollution in the relevant documents. The relevant changes have been annotated. Please review them again in your busy schedule. Thank you!
The author's own theoretical contribution is not significantly reflected in this part.
R: Thank you very much for your valuable comments. After summarizing the existing literature, and in the context of the "double carbon" goal, we will explore how government behavior affects carbon emissions under the economic growth goal, which has important theoretical significance for enriching the relevant theories of sustainable development and practical significance for realizing carbon emission reduction. The above content has been added to the introduction.
It is necessary to demonstrate the measurement method of the capital mismatch index (CAP) and labor mismatch index (LAB).
R: Thank you very much for your valuable comments. We have made a detailed description in the capital mismatch index and the labor capital mismatch index. Please review them in your busy schedule. Thank you!
What’s the relation between economic growth pressure, economic growth targets and carbon emissions. The authors should consider whether there is a threshold effect between economic growth pressure and carbon emissions.
R: Thank you very much for your valuable comments. Under the promotion incentive, the economic growth goals set by local governments are generally higher than their own actual development, stimulating local governments to allocate resources to areas that can effectively achieve economic development and promote economic growth. However, the higher economic growth target will cause economic growth pressure on local governments. At the same time, the higher economic growth target pressure urges local governments to sacrifice the environment for short-term economic growth, resulting in increasingly serious environmental pollution. This paper explores the non-linear impact of economic growth pressure on carbon emissions, and concludes that economic growth pressure has a "U" relationship with carbon emissions, indicating that moderate economic growth pressure has an inhibitory effect on carbon emissions. However, excessive economic growth pressure may distort the economic policy tendency of local governments and officials, may drive local governments to adopt an extensive development model of "emphasizing the economy and neglecting the environment",and even lax environmental governance leads to the imbalance between economic construction and environmental protection (Wang et al., 2016a; Wang et al., 2016b), which has an adverse impact on economic efficiency and carbon emission reduction. This will also have a negative impact on economic efficiency and carbon emission reduction. According to the full sample regression in Table 8 in the article, the regression coefficients of the primary term and the secondary term of the economic growth target are -0.213 and 0.393 respectively, it can be seen that the threshold of moderate economic growth pressure is 0.923 (-0.393/ (2*-0.213)). The ratio of the economic growth rate target announced in the government work report of each province at the beginning of the year to the actual economic growth rate of the previous year is 0.923. When the ratio is less than 0.923, the pressure of economic growth can alleviate carbon emissions. If the ratio is higher than 0.923, the pressure of economic growth will aggravate carbon emissions. The above is the purpose of further exploring the nonlinear relationship between economic growth pressure and carbon emissions. Please review them in your busy schedule. Thank you!
Except for conclusions and policy recommendations, there is more room to focus on the limitations and research outlook.
R: Thank you very much for your valuable comments. In the last part of the article, we listed the research limitations of this paper and the next research direction. On the one hand, this paper focuses on the impact of China's provincial-level economic growth goals and carbon emissions, without in-depth discussion on China's urban level, which may lead to some deviation. Exploring the impact of economic growth goals on carbon emissions based on China's urban level panel data is also a direction for future efforts. On the other hand, this study does not take into account the spatial interaction of economic activities. In fact, air pollution has fluidity and transmission effect, and the practice of economic growth target management in a certain region is likely to affect the environment of adjacent regions. Therefore, it is expected to use the spatial econometric model to explore the spatial agglomeration and spatial spillover effects of economic growth goals on carbon emissions in the future.
Reviewer 2 Report
Does economic growth target stimulate carbon emissions? Evidence from China
REPORT
This paper explores the theoretical mechanisms and transmission paths of economic growth targets affecting carbon emissions from the perspective of economic growth targets, and develops an empirical analysis based on 30 provincial panel data in China from 2003 to 2019.
The author(s) finds that: economic growth targets are positively correlated with carbon emissions under a series of endogeneity and robustness; there are regional heterogeneity, target heterogeneity and structural heterogeneity in the impact of economic growth targets on carbon emissions; after economic growth targets are set, government actions can influence carbon emissions by affecting resource mismatch and industrial restructuring; It is further found that there is a "U" shaped relationship between economic pressure and carbon emissions.
My first concern is that I have to read three pages to see the aim of the paper. It is stated at the end of a very long introduction. I think that the goal should be clearly stated at the beginning.
My second major concern is related with the section on the Theoretical Mechanisms, Section 3.
I think that the Section is not clear at all. It should provide the reader with some intuitions and sounds messages. These should be derived from the existing literature. What I read resembles a nice piece of a newspaper article.
I think that the literature used to justify some sentences should be is not sufficient.
The paper is not very well written, and the authors did not do an excellent job describing the theoretical arguments.
Despite these comments, the research question per se is relevant.
This being said, I think that the paper should be changed. The introduction should have a different structure, Section 3 should be rewritten, and the empirical findings better explained and in a more rigorous way.
Author Response
Dear Editors and Reviewers,
Thank you very much for carefully reading the manuscript. We are very grateful to your helpful and valuable recommendations for further improvements of the paper. We have revised the manuscript according to all of reviewers’ comments and suggestions. All the revised contents have been marked red in the manuscript, and the answers to the reviewers’ comments have been marked dark blue in this response letter. More details are as follows.
My first concern is that I have to read three pages to see the aim of the paper. It is stated at the end of a very long introduction. I think that the goal should be clearly stated at the beginning.
Thank you very much for your valuable comments. We adjusted the structure of the introduction. First, we explained the relationship between economic growth target and the environment in the first paragraph, paving the way for the second paragraph. Then in the second paragraph, it explains the importance of China's carbon emission reduction, and introduces the relationship between economic growth goals and carbon emission reduction. In addition, we have also deleted the redundant parts according to your opinions. We hope the modified results can satisfy you.
My second major concern is related with the section on the Theoretical Mechanisms, Section 3. I think that the Section is not clear at all. It should provide the reader with some intuitions and sounds messages. These should be derived from the existing literature. What I read resembles a nice piece of a newspaper article.
Thank you very much for your valuable comments. The original intention of the theoretical mechanism part is to clarify how economic growth goals affect carbon emissions. Your opinion has given me great inspiration. The theoretical mechanism is actually a part of literature review. If theoretical mechanism is listed in a separate chapter, it will appear that the article review and theoretical mechanism are very weak. Therefore, under comprehensive consideration, we merged the two chapters and added the research of some relevant scholars. Please review them in your busy schedule. Thank you! I look forward to discussing with you again.
Round 2
Reviewer 2 Report
The paper has been improved. I think it can be published as it is. A further check for some typos.